# HAPI2LIBIS (v1.0): A new tool for flexible high resolution radiative transfer computations with libRadtran (version 2.0.5)

Antti Kukkurainen<sup>1,2,\*</sup>, Antti Mikkonen<sup>3,4,\*</sup>, Antti Arola<sup>1</sup>, Antti Lipponen<sup>1</sup>, Ville Kolehmainen<sup>2</sup>, and Neus Sabater<sup>1</sup>

Correspondence: Antti Kukkurainen (antti.kukkurainen@fmi.fi), Antti Mikkonen (antti.mikkonen@fmi.fi)

Abstract. Atmospheric radiative transfer (RT) models are useful tools to increase understanding of the physical interactions and processes occurring in the atmosphere and surface. In the category of free and open-source models, libRadtran is a widely used and versatile package. However, running high-resolution calculations with libRadtran is often tedious since libRadtran does not include all the required information to run line-by-line executions (i.e., resolving individual spectral lines of gases) in a flexible way under specific atmospheric conditions, meaning that external software is required. This poses a problem for a user since generating necessary files for libRadtran requires familiarity with the topic of molecular spectroscopy in addition to knowing how to use the external software which may not be tailored for producing libRadtran compatible files. In this paper, we present HAPI2LIBIS, a compact software tool intended to be used in close connection with libRadtran to enable easy high-resolution calculations. HAPI2LIBIS streamlines the generation of high-resolution molecular absorption data for libRadtran by utilizing the latest version of High-Resolution Transmission Molecular Absorption (HITRAN) database and the HITRAN Application Programming Interface (HAPI) with user-specified spectral ranges and atmospheric profiles. HAPI2LIBIS also stores computed absorption cross-sections to its dynamically constructed lookup-table and includes an option for pressure- and temperature-dependent interpolation in order to accelerate subsequent simulations.

#### 1 Introduction

Atmospheric radiative transfer (RT) models are crucial when investigating the many interactions between electromagnetic radiation and the Earth's atmosphere and surface. RT models are used to simulate the transfer of solar and thermal radiation in the atmosphere and are widely used in Earth Observation research. These models are essential for multiple applications, from instrument calibration (Govaerts and Clerici, 2004) and simulation (Vicent et al., 2016) to atmospheric physics studies covering cloud (Alexandrov et al., 2012), aerosol (Dubovik and King, 2000; Hasekamp and Landgraf, 2005; Randles et al., 2013), volcanic ash (Kylling et al., 2015; Sellitto et al., 2016), gas and trace gas remote sensing fields (Clough et al., 1992; Schwaerzel et al., 2020), as well as for the development of algorithms dedicated to the atmospheric correction process of

<sup>&</sup>lt;sup>1</sup>Finnish Meteorological Institute, Atmospheric Research Centre of Eastern Finland, Kuopio, Finland

<sup>&</sup>lt;sup>2</sup>University of Eastern Finland, Department of Technical Physics, Kuopio, Finland

<sup>&</sup>lt;sup>3</sup>Finnish Meteorological Institute, Earth Observation Research, Helsinki, Finland

<sup>&</sup>lt;sup>4</sup>University of Helsinki, Department of Mathematics and Statistics, Helsinki, Finland

<sup>\*</sup>These authors contributed equally to this work.

airborne or satellite images (Richter and Schläpfer, 2013; Ju et al., 2012). One field of research where RT models are used extensively is satellite remote sensing. Satellite remote sensing provides valuable information about the Earth's land surface, ocean, vegetation, and atmosphere (Donlon et al., 2012). Since RT models are used to model and simulate satellite observations (Tenjo et al., 2017; Lajas et al., 2008), the significant increase in the spectral resolution of spectrometers in many planned satellite missions, such as the FLuorescence EXplorer (FLEX) (Drusch et al., 2017) and the Copernicus Anthropogenic Carbon Dioxide Monitoring (CO2M) (Sierk et al., 2021) missions, imposes higher demands on current RT models.

Many RT models have been developed over the past few decades, gradually increasing their modeling capabilities and their complexity over time. Some of the most well-known models include the MODerate resolution atmospheric TRANsmission (MODTRAN) (Berk et al., 2014), the Second Simulation of a Satellite Signal in the Solar Spectrum (6S) (Vermote et al., 1997), the Matrix Operator MOdel (MOMO) RT code (Fell and Fischer, 2001), the Doubling-Adding KNMI (DAK) model (de Haan et al., 1987), the SCIATRAN software package (Rozanov et al., 2014), the Radiative Transfer for TOVS (RTTOV) model (Saunders et al., 2018), the line-by-line radiative transfer model (LBLRTM) (Clough et al., 2005), or the more recently developed Eradiate project (https://www.eradiate.eu/site/). Among the open-source category of RT models, one widely used and extremely versatile RT model is the uvspec included in the library for radiative transfer (libRadtran) package (Mayer and Kylling, 2005; Emde et al., 2016).

The libRadtran software package consists of a set of programs to carry out RT calculations in the solar and thermal spectral regions. Some of the benefits of libRadtran compared to many other RT codes are that it is free and open-source, it includes polarization, several molecular absorption parameterizations, a collection of RT solvers, a database of cloud and aerosol models, and it has an extensive and clear documentation (https://www.libradtran.org/doc/libRadtran.pdf). While libRadtran offers a comprehensive suite of capabilities for RT simulations, its usability and, therefore, flexibility are more limited when the user wishes to run high spectral resolution simulations. By default, libRadtran relies on band parameterizations, mainly the representative wavelengths absorption parameterization (REPTRAN) (Gasteiger et al., 2014), when computing molecular absorption by atmospheric gases. The REPTRAN parameterization includes three different bandwidths, which are defined at a coarse resolution option of 15 cm<sup>-1</sup>, a medium option of 5 cm<sup>-1</sup>, and a fine option of 1 cm<sup>-1</sup> width. While REPTRAN satisfies the resolution requirements for many applications, it is not suitable for high-resolution simulations (sub 1 cm<sup>-1</sup> spectral resolution) due to the inability to discern subtle absorption line differences. By using a user-defined molecular absorption data file that is computed with a line-by-line model, i.e. highly resolved spectral calculations, and is compatible with libRadtran, it would be possible to increase the desired spectral resolution of libRadtran RT calculations. Unfortunately, libRadtran does not include a line-by-line model by default. To overcome this limitation, the download section of the libRadtran website offers pre-computed line-by-line absorption data files covering the spectral range from 500 nm to 100 microns for the six standard atmosphere files defined in Anderson et al. (1986). These absorption datasets were computed using the RT code GENLN2 (Edwards, 1992) with molecular spectroscopic data from the High-Resolution Transmission Molecular Absorption (HITRAN) database (Gordon et al., 2022) using data from the older 1996 version (Rothman et al., 1998). Thus, the potentially outdated nature of these absorption data files derived from old HITRAN versions and available only for six pre-defined models of atmosphere limits the potential simulation accuracy to be achieved. Besides the possible accuracy limitation, the generation and use of these auxiliary files can make running high spectral resolution simulations with libRadtran an arduous task.

In this context, several programs have been developed to enable high-resolution calculations with libRadtran. For example, the widely used Atmospheric Radiative Transfer Simulator (ARTS) model (Buehler et al., 2005; Eriksson et al., 2011; Buehler et al., 2018), which is a comprehensive tool used for atmospheric radiative transfer applications, can be used to compute molecular absorption files for libRadtran. Another tool available for this purpose is the PYthon scripts for Computational ATmospheric Spectroscopy (Py4CAtS) (Schreier et al., 2019). However, these tools might be cumbersome for new and occasional users due to the extensive options provided by these sophisticated RT codes, the formalism required in input and output files, and the possible requirement for users to download beforehand the spectroscopic data from HITRAN or from other spectroscopic data sources.

To make libRadtran a more versatile program for high spectral resolution RT calculations by easing the simulation process, including high spectral resolution data, we present HAPI2LIBIS. HAPI2LIBIS is a Python-based software that uses the HITRAN Application Programming Interface (HAPI) (Kochanov et al., 2016) as its core to compute molecular absorption files that are compatible with libRadtran. HAPI is a software framework that enables users to access, download, and work with the latest data from the HITRAN database (https://hitran.org/hapi/) by providing a standardized and convenient way to extract spectral molecular data, thus increasing the functionality of the HITRAN database. In addition to HITRAN data, HAPI2LIBIS can utilize measured absorption cross-section data files that are included in libRadtran or provided by the user. These files can be used to complement some gas absorption calculations where HITRAN data is unavailable. The main purpose of HAPI2LIBIS is to enable high-resolution RT calculations with libRadtran for users that require higher spectral resolution than REPTRAN can provide (i.e., finer than 1 cm<sup>-1</sup>) and for users who want to use the latest HITRAN data for line-by-line computations with different atmospheric gas constituents and profiles. These capabilities are particularly valuable for studies involving detailed molecular spectroscopy, where accurate representation of absorption lines is critical, as well as for high-resolution instrument development or characterization studies. While the HITRAN database contains data on absorption cross-sections, collision induced absorption and the water vapor continuum absorption, the current focus of the HAPI2LIBIS is to obtain the line-by-line data from HITRAN and compute the cross-sections locally.

We have also included in HAPI2LIBIS the possibility of utilizing irregular interpolation methods to provide significant computational speed-ups without sacrificing too much accuracy. This option can be highly valuable, for example, when multiple runs with similar atmospheric conditions must be modelled since previously computed data can be reused so that the introduced interpolation error is minimal compared to the other error sources, such as sensor noise. Therefore HAPI2LIBIS could be used as part of a retrieval algorithm when the forward model requires line-by-line calculations. HAPI2LIBIS could act as a replacement for precomputed spectral absorption tables, such as ABSCO (Payne et al., 2020) in total column of CO<sub>2</sub> retrievals, or it could conveniently complement a retrieval algorithm where gas absorption cross-sections are computed as a part of the retrieval algorithm, such as CH<sub>4</sub> profile retrieval SWIRLAB (Tukiainen et al., 2016). As HAPI2LIBIS is used before an RT model is run, it can be utilized simultaneously with spectral RT optimization methods, such as correlated-k (Kato et al., 1999), low-streams interpolation (O'Dell, 2010), or machine-learning based approaches (e.g. Su et al. (2023); Mauceri et al. (2022)).

The remaining of this work is structured as follows: Section 2 provides a detailed overview of the HAPI2LIBIS software package and its use together with libRadtran. Section 3 details, with practical examples, how to make use of HAPI2LIBIS and assess its performance against simulations done in libRadtran using REPTRAN. Additionally, Sect. 3 presents an example of the HAPI2LIBIS cross-section interpolation feature that is included to significantly speed up the computations between similar pre-computed atmospheric files. Finally, conclusions are provided in Sect. 4.

#### 2 HAPI2LIBIS: connecting HITRAN, HAPI, and libRadtran

#### 2.1 Installation and usage

The requirements to run HAPI2LIBIS are Python 3 with NumPy (Harris et al., 2020), SciPy (Virtanen et al., 2020), PyYAML, and NetCDF (Rew and Davis, 1990; Brown et al., 1993) modules. Additionally, an internet connection is required to access the latest HITRAN data. The recommended operating systems to run HAPI2LIBIS are GNU/Linux and macOS systems. Python knowledge is recommended but not required to run HAPI2LIBIS successfully. It is possible to run HAPI2LIBIS without libRadtran installation, but it is not recommended since it would severely limit the functionality of HAPI2LIBIS.

HAPI2LIBIS software can be downloaded from GitHub (https://github.com/amikko/hapi2libis) with MIT licence. The software consists of a Python file called hapi2libis.py and a human-readable text settings file written in yaml. HAPI2LIBIS also requires the hapi.py file, which is obtainable from GitHub (https://github.com/hitranonline/hapi) and from the HITRAN webpage (https://hitran.org/hapi/). The easiest way to install HAPI2LIBIS is to place both of the files, i.e., hapi2libis.py and hapi.py, into the same folder.

HAPI2LIBIS can be operated from the command-line with the command python hapi2libis.py config.yaml, where config.yaml contains all the settings required by HAPI2LIBIS. The settings file contains a section for basic options that the user should edit for their local specification and simulation needs. The settings file also contains a section for advanced options, including features for advanced users and for those who might wish to edit the hapi2libis.py script to be more suitable for their specific needs. The basic options are listed in Table 1.

Before running HAPI2LIBIS for the first time, the user should modify the settings file. The minimum recommended settings that the user should modify to successfully run HAPI2LIBIS is the file path pointing to libRadtran folder (e.g. libradtranpath: '/home/user/bin/libRadtran-2.0.5'), the desired atmosphere file from the ones provided by libRadtran (Anderson et al., 1986), the wavelength range for the computation, and the desired resolution for the output molecular absorption file.

The general workflow of HAPI2LIBIS is presented in Fig. 1. In the initialization phase, the yaml settings file is read following the download of the latest spectroscopy information from the HITRAN database for the required gases in the specified spectral range. Based on the downloaded gases, HAPI2LIBIS will initiate the main computation loop where the absorption coefficients will be computed for every atmospheric layer defined by the model of atmosphere selected from the libRadtran included defaults by Anderson et al. (1986) or defined by the user. Finally, the molecular absorption file (input for libRadtran) is generated.

**Table 1.** Description of the basic parameters in the settings file.

| Parameter name  | Example                       | Description                                            |
|-----------------|-------------------------------|--------------------------------------------------------|
| libradtranpath  | '/home/user/libRadtran-2.0.5' | Location of the libRadtran folder.                     |
| wl_range_nm     | [630, 635]                    | Desired wavelength range in nanometers.                |
| nu_resolution   | 0.01                          | HAPI resolution in wavenumbers.                        |
| atmosphere_id   | 1                             | Number id for atmosphere file.                         |
| name_suffix     | 'test_file'                   | Name suffix for the created molecular absorption file. |
| selected_gases  | ['O2', 'H2O', 'CO2']          | List of gases to use.                                  |
| function_choice | 'Voigt                        | Broadening function.                                   |
| use_xsec_db     | True                          | Use irregular interpolation.                           |

**Figure 1.** Flowchart visualizing the general phases of HAPI2LIBIS. The orange-colored shape for the settings file denotes the start of the flowchart, while the green-colored shapes show the regular flow. The red-colored shapes show the processes that require using the advanced options. The blue gray shape in the middle shows where is the option to speed up computations if the irregular interpolation is activated. Finally, the orange shapes at the end of the flowchart (top right) denote the output files of HAPI2LIBIS.

#### 2.2 The main computation sequence

The easiest way to run HAPI2LIBIS is to use one of the six standard atmosphere files that are included in libRadtran. These files consist of rows and columns, each row of which corresponds to an atmospheric level, while the columns correspond to different parameters. The parameters are altitude above sea level, pressure, temperature, air concentration, and different gas concentrations. The gases included in these files are ozone  $(O_3)$ , oxygen  $(O_2)$ , water vapour  $(H_2O)$ , carbon dioxide  $(CO_2)$ , and nitrogen dioxide  $(NO_2)$ . Additionally, auxiliary gases from the US-standard atmosphere can be used and their concentration profiles are included in separate files in libRadtran. These are methane  $(CH_4)$ , nitrous oxide  $(N_2O)$ , carbon monoxide (CO), and nitrogen  $(N_2)$ . However, for advanced users, HAPI2LIBIS can also read custom atmosphere files that are constructed similarly to the standard atmosphere files provided that the file contains columns for altitude (km), pressure (hPa), temperature (K), and number concentration of air  $(cm^{-3})$ . The additional columns should have the profiles for user-specified gases in the same units as the number concentration of air  $(cm^{-3})$ . The column order of the user-specified gases is not fixed, but the gas concentration should be detailed in rows in altitude top-down order, and the units must be the same as in the standard files. For available gases to use in computations, we refer to the HITRAN website (https://hitran.org/lbl/).

HAPI2LIBIS will compute absorption cross-sections of each of the gases specified in the settings file using the pressures, temperatures, and gas concentrations defined in the atmosphere file for each of the atmospheric layers. Since the atmosphere file consists of n number of levels, HAPI2LIBIS will compute by default the properties of n-1 atmospheric layers where one atmospheric layer consists of two consecutive rows in the atmosphere file, as is the atmospheric definition formalism in libRadtran. This process is referred to as the main loop (see Fig. 1).

The first step in the main loop is computing the layer altitude and thickness, where the user can choose in the settings file if the atmospheric parameters should be interpolated to the mean altitude of the layer. Without altitude interpolation, HAPI2LIBIS uses the lower altitude row values from the atmosphere file to compute the layer properties. For example, a layer from 0 km to 1 km will use the values from the row corresponding to 0 km, while a layer from 1 km to 2 km will use the values from the row corresponding to 1 km and so forth for the rest of the layers. With the altitude interpolation option enabled, HAPI2LIBIS assumes that the row values in the atmosphere file correspond to the values at the given altitude. Therefore, to compute the layer properties HAPI2LIBIS will interpolate the values in the atmosphere file to the layer mean altitude. Compared to the previous example, now a layer from 0 km to 1 km will use the interpolated values corresponding to altitude 0.5 km, while a layer from 1 km to 2 km will use the interpolated values corresponding to altitude 1.5 km and so forth for the rest of the layers. The option to control this is the parameter interpolate\_to\_layer\_midpoint in the settings file, where value True means that the interpolated values are used.

Once the layer properties for layer l are computed, HAPI2LIBIS will move to the second step in the main loop and compute the mixing ratios for every used gas using equation

$$\xi_{\text{gas}}(l) = \frac{c_{\text{gas}}(l)}{c_{\text{air}}(l)},$$
 (1)

where  $\xi_{\rm gas}$  denotes the mixing ratio of a gas in layer l,  $c_{\rm gas}$  denotes the concentration of a gas in layer l, and  $c_{\rm air}$  denotes the concentration of air in layer l. These mixing ratios are later used to communicate to HAPI the gas mixture composition, which affects the air- and self-broadening parameters during the computation of absorption cross-sections.

In the third step of the main loop, HAPI2LIBIS will compute the absorption cross-sections for every gas used. This step is usually the slowest one where the majority of the total computation time will be spent. The cross-sections are computed in the wavelength range specified in the settings file wl\_range\_nm parameter with a wavenumber step option nu\_resolution. The broadening function HAPI2LIBIS uses can be set in the settings file where the possible functions to use are listed in the HAPI manual (https://hitran.org/static/hapi/hapi\_manual.pdf). The function choices are Voigt profile (Voigt), Lorentzian profile (Lorentz), speed-dependent Voigt profile (SDVoigt), Doppler profile (Doppler), and Hartmann-Tran profile (HT). The broadening function accounts for air- and self-broadening where air-broadening refers to the widening of spectral lines caused by collisions between the absorbing gas and surrounding atmosphere, while self-broadening refers to widening due to collisions between molecules of the same gas.

HAPI2LIBIS will utilize HAPI to compute cross-sections for gases downloaded from HITRAN in the initialization phase. The cross-sections are computed one by one, where the HAPI's diluent self parameter is set to the mixing ratio of the gas being computed, while the diluent air is set to the difference between unity and the mixing ratio of the gas being computed.

From the computed cross-sections, HAPI2LIBIS will compute the gas mixture's total absorption coefficient and use that to compute the optical depth of the atmospheric layer l. The total absorption coefficient  $\sigma_{abs}$  is computed with equation

$$\sigma_{\rm abs}(l) = \sum_{\rm gas} Q_{\rm gas}(l) \cdot c_{\rm gas}(l), \tag{2}$$

where  $Q_{\rm gas}$  is the computed absorption cross-section for a gas. Now, the optical depth of the layer l can be computed with the equation

$$\tau(l) = \sigma_{\text{abs}}(l) \cdot z(l), \tag{3}$$

where  $\tau$  denotes the optical depth of the layer and z denotes the layer thickness. Finally, HAPI2libis will save the computed values in a netCDF file specified by hapi2libis\_mol\_tau\_file.nc, which libRadtran is able to use with libRadtran input option mol\_tau\_file abs hapi2libis\_mol\_tau\_file.nc.

#### 180 2.3 Storing calculated absorption cross-sections and their irregular grid interpolation

The calculated absorption cross-sections are stored by default in a NetCDF file for a particular gas, wavelength band, and wavenumber resolution. NetCDF is a widely used data format due to its ability to efficiently store large datasets with multi-dimensional arrays. Utilizing NetCDF files allows HAPI2LIBIS to save and retrieve computed absorption cross-sections efficiently, minimizing storage requirements while ensuring data integrity. The file can store the absorption cross-sections for different temperature (T), pressure (p), and gas partial pressure combinations  $(p_g)$ . To further reduce the computational weight of absorption cross-section calculations, HAPI2LIBIS enables interpolation in the  $(T, p, p_g)$  parameter space. Two irregular interpolation schemes are available: nearest neighbour and bounding-box interpolation.

Both of these two interpolation methods depend on a concept of metric (distance function) in the  $(T, p, p_g)$  space. The metric used in HAPI2LIBIS to compute the distance  $d_w$  between two points  $r_i = (T_i, p_i, p_{gi})$  and  $r_j = (T_j, p_j, p_{gj})$  is

$$d_w(r_i, r_j) = (r_j - r_i)^{\mathrm{T}} W(r_j - r_i),$$
 (4)

where  $W \in \mathbb{R}^{3 \times 3}$  and  $W = \operatorname{diag}(w_1, w_2, w_3)$ . The values  $w_1, w_2$  and  $w_3$  can be different for each gas, wavenumber region, spectral resolution, broadening function as well as pressures and temperatures used. For a particular dataset consisting of points  $r_i = (T_i, p_i, p_{qi}), i = 0, 1, ..., n$ , the weighting matrix W is

$$W = \begin{bmatrix} (\max\{T_i\} - \min\{T_i\})^{-2} & 0 & 0\\ 0 & (\max\{p_i\} - \min\{p_i\})^{-2} & 0\\ 0 & 0 & (\max\{p_{gi}\} - \min\{p_{gi}\})^{-2} \end{bmatrix}.$$
 (5)

To interpolate point  $r=(T,p,p_g)$ , an inverse distance weighting method (Franke, 1982) is used. The described method is modified by not using all the points in the dataset, but only two:  $r_i$  and  $r_j$ . These two points are selected from a subset  $C=\{r_i|i\in 0,1,...,n \land d_w(r,r_i)< d_c\}$ , where  $d_c$  threshold value is selected such that  $|C|=n_c$ , for which  $n_c$  is chosen arbitrarily beforehand. In other words, these two points are picked from  $n_c$  closest points to r. The two points must also fulfill the following criteria:

- 1.  $\min(T_i, T_j) < T < \max(T_i, T_j),$
  - 2.  $\min(p_i, p_j)$
  - 3.  $\min(p_{qi}, p_{qj}) < p_q < \max(p_{qi}, p_{qj}),$
  - 4.  $v(r_i, r_j) < v_{\text{max}}$ , and
  - 5.  $v(r_i, r_j) <= v(r'_i, r'_j) \forall r'_i, r'_j \in C$ ,

where

$$v = \det(\operatorname{diag}(W[|T_i - T_j|, |p_i - p_j|, |p_{gi} - p_{gj}|]^{\mathrm{T}}))$$
(6)

is the volume of the box defined by the points  $r_i$  and  $r_j$  in the  $d_w$  metric. More simply put, the interpolant must be contained in the box defined by  $r_i$  and  $r_j$ , the box volume must be less than a predetermined threshold value, and the selected box will be the smallest box in the set C.

With suitable  $r_i$  and  $r_j$ , the interpolated absorption cross-section  $Q_{gas}$  at the point r is

$$Q_{\text{gas}}(r) = \frac{Q_{\text{gas}}(r_i)d_w(r, r_j) + Q_{\text{gas}}(r_j)d_w(r, r_i)}{d_w(r, r_i) + d_w(r, r_i)}.$$
(7)

#### 2.4 Extending HITRAN with auxiliary tables from libRadtran

The HITRAN database does not cover all the wavelengths where significant molecular absorption is encountered in the atmosphere. For example, the Chappuis band of  $O_3$ , which covers the wavelength range between 400 and 650 nm, cannot be downloaded from the HITRAN database. LibRadtran manages this by including several measured absorption cross-section files below 1130 nm for  $O_3$ ,  $NO_2$ , and for  $O_2$ - $O_2$  collision pairs.

To make HAPI2LIBIS comparable to the highest resolution option of REPTRAN, we have included in HAPI2LIBIS the possibility of using some of the measured absorption cross-section files included in libRadtran (Emde et al., 2016). The measured absorption cross-section table values are used as per the instructions in the file. This means that they may lack pressure and temperature effects. The values are interpolated to the requested wavelength range and resolution, meaning that no line-broadening effects are applied. For O<sub>3</sub>, the available files are molina (Molina and Molina, 1986), bass\_and\_paur (Bass and Paur, 1985), daumount (Daumont et al., 1992), and bogumil (Bogumil et al., 2003). For NO<sub>2</sub> the available files are burrows (Burrows et al., 1998), and bogumil (Bogumil et al., 2003). For O<sub>2</sub>-O<sub>2</sub>, the only available file is the greenblatt (Greenblatt et al., 1990).

#### 225 3 Model comparison results

## 3.1 Comparison of libRadtran simulations

For this section's results, the libRadtran simulation specifications are fixed to the following defaults unless otherwise stated. For the measurement geometry, the solar zenith angle and the surface albedo values were set to 35 degrees and 0.3, respectively. The atmosphere was chosen as the Mid-latitude Summer. The atmosphere contains default type aerosols according to Shettle (1990), which correspond to spring-summer conditions with a visibility of 50 km and rural type aerosols in the lower 2 km and background type aerosols above 2 km in the atmosphere. These simulation parameters were chosen since they represent typical mid-latitude summer conditions, which are commonly used as a benchmark in atmospheric radiative transfer studies.

The output spectral resolution in libRadtran is usually defined by the chosen extraterrestrial spectrum. We decided to use the full resolution spectrum according to Kurucz (Kurucz, 1992), which can be downloaded from the libRadtran webpage. Concerning HAPI2LIBIS, we used every available isotope for the downloaded HITRAN gases and the default auxiliary tables for  $O_3$ ,  $NO_2$ , and  $O_2$ - $O_2$  that are molina, burrows, and greenblatt, respectively. The broadening function was set as Voigt and the wavenumber resolution was  $0.01 \text{ cm}^{-1}$ .

Figure 2 shows libRadtran simulated direct transmission results when using molecular absorption option REPTRAN fine (1 cm<sup>-1</sup>) and HAPI2LIBIS molecular absorption file with full (a) and a simulated instrument (b) resolution. The instrument spectra are computed by convolving the full resolution spectra with a triangular slit function with a full width at half maximum (FWHM) of 0.3 nm, which represents a typical high-resolution instrument. The trace gases that were included in the simulations were O<sub>3</sub>, O<sub>2</sub>, H<sub>2</sub>O, CO<sub>2</sub>, NO<sub>2</sub>, CH<sub>4</sub>, N<sub>2</sub>O, CO, N<sub>2</sub>, and O<sub>2</sub>-O<sub>2</sub>. The figure shows the higher resolution of HAPI2LIBIS compared to REPTRAN, which is as expected since REPTRAN is not a line-by-line method and has a greater priority for

simulation speed over accuracy. The differences in spectral resolution between HAPI2LIBIS and REPTRAN are particularly noticeable in the 2000–2500 nm range where HAPI2LIBIS shows many of the finer absorption lines that are missing in the REPTRAN data. For easier comparison, Figs. A1 and A2 in the Appendix A show difference plots of the results shown in Fig. 2. The libRadtran simulation to generate the spectra shown in Fig. 2 took around 13 minutes with REPTRAN and around 92 minutes with HAPI2LIBIS (single thread on Intel<sup>®</sup> Xeon<sup>®</sup> Gold 6230 CPU, 2.1 GHz, 64 GB memory).

Figure 3 shows simulated top-of-atmosphere (TOA) radiance results with full (a) and instrument (b) resolution using libRadtran paired with REPTRAN fine, HAPI2LIBIS molecular absorption file, and the line-by-line absorption data that can be downloaded from the libRadtran webpage. The observing instrument has nadir view geometry. The trace gases that are present in the libRadtran line-by-line molecular absorption files are  $O_3$ ,  $O_2$ ,  $H_2O$ ,  $CO_2$ ,  $CH_4$ ,  $N_2O$ , CO, and  $N_2$  but for this specific example, the trace gases that were included in the simulation were  $O_2$ ,  $H_2O$ , and  $CO_2$ . These gases were selected since they all appear in the HITRAN database for the requested wavelength range meaning that the results are comparable. The figure shows that all three methods show major absorption lines at the same wavelengths, which is as expected. The figure also shows that REPTRAN has significantly smaller absorption spikes compared to the other two methods due to the coarser spectral resolution, while HAPI2LIBIS and line-by-line files provide similar results. The difference in the height of the spikes between HAPI2LIBIS and line-by-line files can be attributed to the different versions of used HITRAN (2020 and 1996, respectively) or due to some possible differences in the used line broadening function. The figure also shows that there is no substantial difference between the convolved lower resolution spectra despite REPTRAN missing many of the absorption spikes; however, for high resolution instrument data studies, these differences are significant.

Figure 4 shows the simulated TOA radiance results with full (a) and instrument (b) resolution when we include  $O_3$  to the simulations that were presented in Fig. 3. Since the libRadtran line-by-line data files are based only on HITRAN data, which does not include  $O_3$  on the chosen wavelengths, we can clearly see the difference in the TOA radiance between REPTRAN and HAPI2LIBIS compared to the libRadtran line-by-line data.

# 3.2 Comparison of interpolation methods

As a demonstration of the accuracy and speed of the irregular interpolation method, we processed 40 different atmospheric profiles used as priors in the Total Carbon Column Observation Network (TCCON) retrieval (Wunch et al., 2011). The prior profiles used are from the Sodankylä TCCON (Kivi et al., 2022) from 1 March 2023 to 11 April 2023. Using the provided pressure, temperature profiles, as well as mixing profiles for CO<sub>2</sub>, CH<sub>4</sub> and H<sub>2</sub>O, the absorption cross-sections were computed in the 1600 – 1700 nm wavelength range. In that range, 36765 discrete wavenumbers were used for calculating Voigt line shapes of 17210, 12858, and 3507 absorption lines for CO<sub>2</sub>, CH<sub>4</sub>, and H<sub>2</sub>O, respectively. Each of the used atmospheres had 50 layers from 0 to 70 km and layer thicknesses from 0.42 to 2.38 km. As an example, Fig. 5 shows the optical densities used in this comparison exercise at different altitudes. To demonstrate HAPI2LIBIS capabilities in the thermal infrared wavelength region, a similar example is presented in Fig. A3.

**Figure 2.** Comparison of simulated direct transmission from Sun to surface at full (a) and instrument (b) resolution with HAPI2LIBIS generated molecular absorption optical depth file against REPTRAN fine molecular absorption parameterization. The instrument response function was a triangular slit function with a full width at half maximum (FWHM) of 0.3 nm.

**Figure 3.** Simulated TOA radiance at full (a) and instrument (b) resolution using REPTRAN, HAPI2LIBIS, and line-by-line molecular absorption data files provided in the libRadtran webpage. The atmospheric simulation includes  $O_2$ ,  $H_2O$ , and  $CO_2$ .

Figure 4. Simulated TOA radiance at full (a) and instrument (b) resolution using REPTRAN, HAPI2LIBIS, and line-by-line molecular absorption data files provided in the libRadtran webpage. Atmospheric The distinction to the results shown in Fig. 3 is that now the atmospheric O<sub>3</sub> is included in the simulations.

Figure 5. Examples of the fully computed total spectral optical densities used in interpolation comparison at different altitudes in the atmosphere from the first processed atmosphere (left) and zoomed in section to highlight the altitude dependence on the line shape (right). Contributions of CO<sub>2</sub>, CH<sub>4</sub> and H<sub>2</sub>O are aggregated together as this is the content of the mol\_abs file created by HAPI2LIBIS.

**Table 2.** The filesizes of the absorption cross-section databases after processing the 40 atmospheres.

| Interpolation method                | CO <sub>2</sub> (Mb) | H <sub>2</sub> O (Mb) | CH <sub>4</sub> (Mb) | total size (Mb) |
|-------------------------------------|----------------------|-----------------------|----------------------|-----------------|
| Nearest neighbour, $v_{max} = 0.01$ | 223.2                | 167.6                 | 226.7                | 617.5           |
| Bounding-box, $v_{max} = 0.001$     | 21.8                 | 16.9                  | 18.7                 | 57.4            |
| Bounding-box, $v_{max} = 0.0001$    | 39.8                 | 26.1                  | 30.3                 | 96.2            |
| Bounding-box, $v_{max} = 0.00001$   | 61.8                 | 37.1                  | 56.7                 | 155.6           |
| Bounding-box, $v_{max} = 0.000001$  | 78.0                 | 47.4                  | 89.3                 | 214.7           |

In Fig. 6, the interpolation speed-ups with different interpolation schemes are compared to fully computing the cross-sections. The speed-up comparison was done on Intel<sup>®</sup> Core<sup>TM</sup> i7-8650U CPU. Additionally, in Table 2, the file sizes of the accumulated databases after processing 40 atmospheres are presented.

The interpolation methods decrease the runtime needed for absorption cross-section computations as is apparent in Fig. 6, increasingly so when more atmospheres are computed. The bounding-box method enables two to twenty times faster computation of the cross-sections, depending on the parameter. The nearest-neighbour is also faster than the full computation, but the runtime reduction is hardly as drastic. The speed-up is still increasing since the runtime curves for interpolators have a downward trend. Since the interpolation methods accumulate the computed cross-sections into the database, it is reasonable

Figure 6. The runtime for processing the absorption cross-sections for each gas at every layer for a single atmosphere as a function of the amount of previously processed atmospheres. FC stands for full computation, NN for nearest-neighbour, and BB for bounding-box. The  $v_{max}$  parameter for NN is 0.01 and for BB it is included in the legend.

that the fewer cases are interpolable, the more need to be computed and stored in the database, which is visible from the Table

2. The total stored cross-section sizes ranging from 57.4 Mb to 617.5 Mb for different interpolation methods show that the fastest method also requires the least disk space.

In Fig. 7, the relative error in the resulting transmittance is presented as a function of the atmospheric index for each interpolation method. The transmittance is computed with the formula

$$T(\lambda) = \exp\left(-\sum_{i=1}^{50} l_i \tau_i(\lambda)\right),\tag{8}$$

where  $l_i$  is the thickness of i'th layer and  $\tau_i(\lambda)$  is the optical density at i'th atmospheric layer. In Fig. 7, the tradeoff of accuracy versus computation speed is visible. Albeit requiring most compute time and disk space, the nearest neighbour (a) performs consistently, with the largest outlier errors being 3% for all the 40 atmospheres. The bounding-box (b–e) methods can have outlier errors up to 50%, but the errors are reduced as more atmospheres are computed. The decrease of  $v_{max}$  from 0.001 to 0.0001 increases the accuracy, but reducing  $v_{max}$  more does not visibly increase accuracy further. The bounding-box method appears to require a burn-in period of about 15 atmospheres before the error distribution becomes somewhat stationary. After the burn-in period, the 3-sigma envelope ranges from 1–2%, signifying that almost all spectral points are accurately interpolated.

**Figure 7.** The relative error across the spectra for each interpolation method as a function of the amount of previously processed atmospheres. The mean relative error is the solid blue line, the 3-sigma envelope (99.7 % of errors closer than mean when assuming normality) is the dashed red line, and all the errors are contained within dotted black line.

The requirement for the burn-in period can be explained by examining Eq. 5, which represents the full range of the data within the database. Before the database is adequately populated, the weighting matrix can cause the interpolation not to be strict enough, which results in higher errors. The large outlier errors can be explained by the linear nature of the bounding-box interpolation. Because the Voigt line shapes themselves are not linear with respect to p, T or  $p_g$ , the linear interpolation can cause some unphysical peak shifts, which can result in large relative errors in transmittance, especially in the presence of fully saturated absorption lines. This can be a problem in all cases where linear interpolation of absorption cross-sections is done. However, in the use cases where spectral observations of specific instruments are simulated, the instrument line shape function can be broad enough to nullify the errors of these rare outliers.

## 4 Conclusions

In this paper, we presented HAPI2LIBIS, a Python-based tool designed to integrate HITRAN data with libRadtran by interfacing HAPI with the aim of enabling easy high-resolution radiative transfer calculations with libRadtran. To achieve this goal, HAPI2LIBIS has been made as interoperable with libRadtran as possible. With HAPI2LIBIS, the user is able to compute molecular absorption cross-sections of various gases for different atmospheric conditions from the ultraviolet to the microwave region.

In HAPI2LIBIS, the process of computing absorption cross-sections benefits from the high precision molecular absorption line data available in the HITRAN database. Unlike some alternative strategies that rely on precomputed tables or band-parameterized models, HAPI2LIBIS calculates absorption cross-sections dynamically based on the gas concentrations in each atmospheric layer, thus improving the accuracy of the simulations.

The use of HAPI2LIBIS can be particularly useful for applications such as high-resolution instrument simulations and studies requiring precise representation of molecular absorption features. The cross-section interpolation database enables rapid processing of a series of atmospheres with only slight errors as well as boosting repeated computational studies by conveniently storing the previously computed cross-sections.

As a final remark, most of the capabilities present in HAPI2LIBIS can be founded in other sophisticated models. For example, both LBLRTM and Py4CAtS can compute absorption cross-sections of gases based on HITRAN data. However, the core design principle of HAPI2LIBIS is to add more functionality to libRadtran while being relatively easy to use. In future HAPI2LIBIS versions, planned updates include implementing parallel computation techniques to the cross-section computation, utilizing more data from HITRAN than only the line-by-line, and the option to use different line broadening functions for different altitudes instead of using only one for the whole profile.

Code and data availability. The current version of HAPI2LIBIS is publicly available on GitHub at (https://github.com/amikko/hapi2libis) under the MIT licence. The exact version of HAPI2LIBIS used to produce the results shown in this paper as well as the input data and scripts

to produce the plots presented in this paper are archived on Zenodo under the DOI 10.5281/zenodo.14673990 (Kukkurainen and Mikkonen, 2025).

# 330 Appendix A: Additional figures

**Figure A1.** Differences in simulated direct transmissions from Sun to surface at full (a) and instrument (b) resolution when utilizing HAPI2LIBIS generated molecular absorption optical depth file and REPTRAN fine molecular absorption parameterization.

**Figure A2.** Histograms of the differences in simulated direct transmissions from Sun to surface at full (a-b) and instrument (c-d) resolution when utilizing HAPI2LIBIS generated molecular absorption optical depth file and REPTRAN fine molecular absorption parameterization. For visual clarity, the vertical axis in panels (a) and (c) is cut.

**Figure A3.** Examples of optical thicknesses computed at different altitudes of the atmosphere by HAPI2LIBIS in the thermal infrared wavelength region. Similarly to Fig. 5, contributions of CO<sub>2</sub> and H<sub>2</sub>O are aggregated together.

Author contributions. AK and AM developed the software, performed the numerical simulations, and wrote the first draft of the manuscript. AA, AL, VK, and NS collaborated on the design of the software, supervised the work, and contributed to the writing and editing of the manuscript.

Competing interests. The authors declare that they have no conflict of interest.

Acknowledgements. This work was supported by the UEF Doctoral Programme in Science, Technology and Computing (SCITECO, dnro 203/03.01.02/2020), by the Research Council of Finland grant No 356937 (ArcticSIF), by the Research Council of Finland flagship of Atmosphere and Climate Competence Center grants No 357904 and 337552 (ACCC), by the Research Council of Finland grant No 331829 (CitySpot), by the Research Council of Finland Flagship of Advanced Mathematics for Sensing, Imaging and Modeling grants No 358944 and 359196 (FAME) and by the Research Council of Finland Centre of Excellence in Inverse Modelling and Imaging grants No 353084 and 353082.

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
