# Peer review of "HAPI2LIBIS (v1.0): A new tool for flexible high resolution radiative transfer computations with libRadtran (version 2.0.5)"

_EGUsphere, 2025_

## Author Response (AR1)

**Authors response to comments**

https://doi.org/10.5194/egusphere-2025-220 Geoscientific Model Development July 25, 2025

We thank the reviewers for all the comments and suggestions. In this response document, we have answered to every question raised by the reviewers. **Bold text** are the comments/suggestions made by the reviewers. The changes we have made to manuscript are displayed in *italics* in this reply.

Reviewer 1 Comments to the Author: This manuscript introduces a software tool HAPI2LIBIS to compute absorption cross section of atmospheric gases based on HITRAN dataset, and can provide input file of absorption cross section for libRadtran radiative transfer computations. One of the features is the irregular interpolation method to interpolate absorption cross section at different temperature, pressure and gas concentration grids. Some comments are as follows.

1. Line-by-line Radiative Transfer Model (LBLRTM, http://rtweb.aer.com/lblrtm.html) can also compute absorption cross section of gases based on HITRAN dataset, and has been widely used in the atmospheric radiation community. The authors should discuss the differences of the HAPI2LIBIS and LBLRTM in terms of gas absorption cross section calculation, and point out what are the specific strengths of the new-developed HAPI2LIBIS.

Response: While LBLRTM also computes the gas absorption cross-sections from the HITRAN spectral line data, HAPI2LIBIS has several strengths and differences compared to it. HAPI2LIBIS is simpler and easier to use, with only two Python code files (one of which is the HITRAN API) and a configuration file, to create customized absorption molecular absorption file for libRadtran. In addition, HAPI2LIBIS automatically downloads the latest available molecular line data from the HITRAN database, so manual absorption line updating and configuration is unnecessary. In HAPI2LIBIS, it is possible to select which line broadening functions are utilized, and the computed absorption cross-sections are stored in a dynamic interpolation tables for reuse. While some of the previously mentioned functionality is present in other models, the primary purpose of HAPI2LIBIS is to extend the functionality of libRadtran meaning the compatibility is guaranteed with up-to-date versions of libRadtran.

We have revised the introduction by adding a reference to the LBLRTM model:

Some of the most well-known models include the MODerate resolution atmospheric TRANsmission (MODTRAN) (Berk et al., 2014), the Second Simulation of a Satellite Signal in the Solar Spectrum (6S) (Vermote et al., 1997), the Matrix Operator MOdel (MOMO) RT code (Fell and Fischer, 2001), the Doubling-Adding KNMI (DAK) model (de Haan et al., 1987), the SCIATRAN software package (Rozanov et al., 2014), the Radiative Transfer for TOVS (RTTOV) model (Saunders et al., 2018), the

line-by-line radiative transfer model (LBLRTM) (Clough et al., 2005), or the more recently developed Eradiate project (https://www.eradiate.eu/site/).

Additionally, we added the following paragraph to the conclusions as an additional motivation for developing the tool:

As a final remark, most of the capabilities present in HAPI2LIBIS can be founded in other sophisticated models. For example, both LBLRTM and Py4CAtS can compute absorption cross-sections of gases based on HITRAN data. However, the core design principle of HAPI2LIBIS is to add more functionality to libRadtran while being relatively easy to use.

**2. The authors should also discuss more about the computational efficiency aspect of the HAPI2LIBIS if used in radiative transfer computation and remote sensing. Will HAPI2LIBIS be used in any remote sensing retrieval algorithms?**

Response: Almost always retrieval algorithms are based on pre-computed tables of spectral radiances or gas absorption cross-sections, so very rarely are line broadening calculations done on-the-fly, so to say. However, the interpolation from the HAPI2LIBIS tables is rapid, as long as sufficient amount of parameter points are added, so in principle it could be used to replace spectral absorption tables, such as ABSCO (https://doi.org/10.1016/j.jqsrt.2020.107217), within a retrieval algorithm. On the other hand, in some retrievals, such as CH4 profile retrieval algorithm SWIRLAB (https://doi.org/10.1002/2015JD024657), the gas absorption cross-sections are computed as a part of the retrieval algorithm, because high spectral and temperature-pressure dependency accuracies are needed. In such retrieval algorithms, HAPI2LIBIS could prove to be practical.

We have expanded the introduction with the following paragraph:

Therefore HAPI2LIBIS could be used as part of a retrieval algorithm when the forward model requires line-by-line calculations. HAPI2LIBIS could act as a replacement for precomputed spectral absorption tables, such as ABSCO (Payne et al., 2020) in total column of CO2 retrievals, or it could conveniently complement a retrieval algorithm where gas absorption cross-sections are computed as a part of the retrieval algorithm, such as CH4 profile retrieval SWIRLAB (Tukiainen et al., 2016). As HAPI2LIBIS is used before an RT model is run, it can be utilized simultaneously with spectral RT optimization methods, such as correlated-k (Kato et al., 1999), low-streams interpolation (O'Dell, 2010), or machine-learning based approaches (e.g. Su et al. (2023); Mauceri et al. (2022)).

Reviewer 2 Comments to the Author: Atmospheric gas absorption has been a long-lasting task for radiative transfer simulations due to the complex absorption lines. Meanwhile, as a widely applied radiative transfer (RM) model, the libRadtran is lack of a high-resolution LBL-based gas absorption model, which clearly limits its applications. This manuscript introduces the new tool for libRadtran, i.e., HAPI2LIBIS (v1.0), for high resolution gas absorption and RT simulations. The new tool is clearly presented and introduced, and fits the GMD well. Considering the wide applicability of libRadtran, the development of such an LBL-based model/tool becomes highly necessary and meaningful, so I would suggest the paper to be accepted for publication after addressing my following concerns.

1. The Abstract briefly introduces the motivation of the new tool, while missed key techniques and contributions of the HAPI2LIBIS. Thus, the Abstract should be substantially improved to indicate the novelty and techniques of the new tool for better understanding.

Response: We initially wrote the abstract to be concise and include only the main purpose/functionality of HAPI2LIBIS. However, we now see that the abstract is lacking necessary information about the techniques presented in the manuscript. Thus, we have extended the abstract with the following paragraph:

HAPI2LIBIS streamlines the generation of high-resolution molecular absorption data for libRadtran by utilizing the latest version of High-Resolution Transmission Molecular Absorption (HITRAN) database and the HITRAN Application Programming Interface (HAPI) with user-specified spectral ranges and atmospheric profiles. HAPI2LIBIS also stores computed absorption cross-sections to its dynamically constructed lookup-table and includes an option for pressure- and temperaturedependent interpolation in order to accelerate subsequent simulations.

2. A traditional LBL model is useful and necessary, while it is computational expensive. Thus, some fast/efficient models have been developed for high spectral resolution simulations, and show great potential for further applications. Gas absorption has been a key part for those fast models. For example, the principle-component method and machine learning techniques have been used for high-resolution gas absorption simulations (see https://doi.org/10.1007/s00376-023-2293-5 and https://doi.org/10.1007/s00376-021-0366-x). Would the authors think that those models be possible to be included in LibRadtran in the future to replace LBL models, and some introductions on the fast models are suggested in the manuscript.

Response: Thank you for the paper suggestions. While these models could be of use for libRadtran, it is up to the libRadtran authors to include them there. Some spectral RT speedup methods are already included in libRadtran, such as correlated-k and pseudo-spectral calculation. It must be noted that HAPI2LIBIS only creates the line-by-line input for libRadtran, whereas these methods are distinct approaches for handling spectral RT.

Including fast, efficient, and accurate models for gas absorption in HAPI2LIBIS is something that could be added in the future. We have added a mention of fast models and their potential in the

manuscript (check response to comment 2 by reviewer 1). For convience, below is the paragraph we added to the introduction.

Therefore HAPI2LIBIS could be used as part of a retrieval algorithm when the forward model requires line-by-line calculations. HAPI2LIBIS could act as a replacement for precomputed spectral absorption tables, such as ABSCO (Payne et al., 2020) in total column of CO2 retrievals, or it could conveniently complement a retrieval algorithm where gas absorption cross-sections are computed as a part of the retrieval algorithm, such as CH4 profile retrieval SWIRLAB (Tukiainen et al., 2016). As HAPI2LIBIS is used before an RT model is run, it can be utilized simultaneously with spectral RT optimization methods, such as correlated-k (Kato et al., 1999), low-streams interpolation (O'Dell, 2010), or machine-learning based approaches (e.g. Su et al. (2023); Mauceri et al. (2022)).

3. Due to the oscillation features of the gas absorption, the comparisons in Figure 2 are not that clear. Thus, the differences between HAPI2LIBIS and REPTRAN are suggested to be shown as well.

Response: To ease comparisons, we have added two different kinds of plots to the Appendix showing the differences between HAPI2LIBIS and REPTRAN, corresponding to the case shown in Figure 2. Figure B1 shows lineplots and Figure B2 shows histograms of the differences, respectively. We have also added the following sentence to the manuscript:

For easier comparison, Figs. A1 and A2 in the Appendix A show difference plots of the results shown in Fig. 2.

For convenience, the added figures are presented below.

Figure A1: Differences in simulated direct transmissions from Sun to surface at full (a) and instrument (b) resolution when utilizing HAPI2LIBIS generated molecular absorption optical depth file and REPTRAN fine molecular absorption parameterization.

Figure A2: Histograms of the differences in simulated direct transmissions from Sun to surface at full (a-b) and instrument (c-d) resolution when utilizing HAPI2LIBIS generated molecular absorption optical depth file and REPTRAN fine molecular absorption parameterization. For visual clarity, the vertical axis in panels (a) and (c) is cut.

**4. Figure 4 compares LBL, HAPI2LIBIS and REPTRAN results, while O3 absorption is not considered in LBL simulations. Such a comparison becomes unfair, and could be possible to confuse readers. A fair comparison without O3 is suggested to be added or to replace the current results.**

Response: Fair comparison is presented in Figure 3. The purpose of Figure 4 is to demonstrate the effect of O3 in the wavelength range 630–635 nm and highlight the capability of HAPI2LIBIS to include new gas information, as in this particular case where the O3 absorption data is missing in HITRAN for this wavelength range. We have rephrased the caption in Figure 4 to be clearer and it

**now reads as:**

Figure 4. Simulated TOA radiance at full (a) and instrument (b) resolution using REPTRAN, HAPI2LIBIS, and line-by-line molecular absorption data files provided in the libRadtran webpage. The distinction to the results shown in Fig. 3 is that now the atmospheric O3 is included in the simulations.

**5. Again, Figure 5 is not clearly shown due to the absorption feature.**

Response: Figure 5 was clarified to make the altitude-dependence of the absorption lines more visible. For convenience, the figure is presented below:

Figure 5: Examples of the fully computed total spectral optical densities used in interpolation comparison at different altitudes in the atmosphere from the first processed atmosphere (left) and zoomed in section to highlight the altitude dependence on the line shape (right). Contributions of  $CO_2$ ,  $CH_4$  and  $H_2O$  are aggregated together as this is the content of the mol\_abs file created by HAPI2LIBIS.

**6. How are the continuum absorptions considered in the model? It is not mentioned that much in the manuscript.**

Response: Despite the water vapour continuum absorption being available on HITRAN, we have decided to focus here on the distinct gas absorption lines. Continuum absorption could be included later on as part of HAPI2LIBIS, but for example, the MT\_CKD\_H2O continuum absorption model is released under a non-free license and would thus not be straightforward to include in HAPI2LIBIS. We have added the following sentence in the introduction to clarify these current limitations in HAPI2LIBIS:

While the HITRAN database contains data on absorption cross-sections, collision induced absorption

and the water vapor continuum absorption, the current focus of the HAPI2LIBIS is to obtain the lineby-line data from HITRAN and compute the cross-sections locally.

Additionally, we rephrased the conclusions to clarify that we will investigate the possibility of including additional features to HAPI2LIBIS. The revised sentence now reads as:

In future HAPI2LIBIS versions, planned updates include implementing parallel computation techniques to the cross-section computation, utilizing more data from HITRAN than only the line-by-line, and the option to use different line broadening functions for different altitudes instead of using only one for the whole profile.

7. The manuscript mainly shows results over the near-infrared and solar bands. However, gas absorption and high-resolutions are also widely used for infrared bands. Would some examples over the longwave-infrared bands be presented and discussed as well?

Response: Example image of the optical thickness of different gases in thermal infrared wavelength band was included in the appendix and the following sentence was added.

To demonstrate HAPI2LIBIS capabilities in the thermal infrared wavelength region, a similar example is presented in Fig. A3.

For convenience, the added figure is presented below.

Figure A3: Examples of optical thicknesses computed at different altitudes of the atmosphere by HAPI2LIBIS in the thermal infrared wavelength region. Similarly to Fig. 5, contributions of  $CO_2$  and  $H_2O$  are aggregated together.

**Other changes to the manuscript:**

1. The colors in the flowchart shown in Figure 1 was changed to be more colorblind safe. The updated figure and caption is shown below.

Figure 1: Flowchart visualizing the general phases of HAPI2LIBIS. The orange-colored shape for the settings file denotes the start of the flowchart, while the green-colored shapes show the regular flow. The blue-colored shapes show the processes that require using the advanced options. The gray shape in the middle shows where is the option to speed up computations if the irregular interpolation is activated. Finally, the orange shapes at the end of the flowchart (top right) denote the output files of HAPI2LIBIS.